# Preparation and characterization of coal-based graphite from Huyan mountain anthracite by high-temperature simulation

Gaojian Chen[1], Daiyong Cao [2]*, Fengchu Liao[1], Hongsheng He[1], Anmin Wang[2], Simai Peng[1]

**1** Hunan Provincial Key Laboratory of Geochemical Processes and Resource Environmental Effects, Changsha, China, **2** College of Geoscience & Surveying Engineering, China University of Mining & Technology, Beijing, China

* cdy@cumtb.edu.cn

## Abstract

This study systematically investigates the graphitization behavior of high-metamorphic anthracite from Huyan Mountain (Shanxi, China) under extreme thermal conditions (2100–3000 °C) through integrated experimental and micro-structural analyses. Acid-washed and demineralized coal samples, with or without $Fe_2O_3$ catalyst, were subjected to controlled thermal treatment to evaluate structural evolution and catalytic effects. X-ray diffraction (XRD) analysis identifies a critical graphitization threshold at $d_{002}$ = 0.3368 nm, beyond which interlayer spacing ceases reduction despite continued lattice refinement. Below 2700 °C, $Fe_2O_3$ catalysis significantly accelerates aromatic layer stacking and in-plane defect healing, advancing graphitization process. Post-threshold stabilization (≥2700 °C), both catalytic and non-catalytic systems exhibit analogous $d_{002}$ stagnation, yet high-resolution transmission electron microscopy (HRTEM) reveals persistent structural ordering, including increased carbon layer stacking (up to 10 layers) and reduced edge defects. Comparative scanning electron microscopy (SEM) demonstrates enhanced flake alignment and interlayer compactness in catalyzed samples. These findings highlight intrinsic limitations in coal-derived graphite synthesis, emphasizing precursor composition as a decisive factor in graphitization potential. The work provides critical insights into graphitization mechanisms and constraints for artificial coal-based graphite production.

## 1. Introduction

Coal-based graphite, which is the final unit of coal metamorphic evolution, is also an important part of cryptocrystalline graphite [1]. It not only has excellent properties such as high temperature resistance, high electrical conductivity, and corrosion resistance [2], but also can be used as a raw material for graphene preparation [3, 4].

**Data availability statement:** All relevant data are within the paper and its Supporting Information files.

**Funding:** This study is financially supported by the National Natural Science Foundation of China (Grant No. 42072197), the Fundamental Research Funds for the Central Universities (2023ZKPYDC07), the Key Research and Development Program of Hunan Province (Grant No. 2022SK2090), and the Major Scientific and Technological Research Project of the Department of Natural Resources of Hunan Province (Xiang Zi Zi Ke 2022[02]).

**Competing interests:** The authors have declared that no competing interests exist.

Therefore, it has broad development prospects in high-tech fields such as metallurgy, military industry, aerospace, and precision electronic components [5].

The evolution of coal-based graphite is governed by both internal and external factors. The internal factors, consisting of organic components and inorganic minerals, serve as the bedrock of the coal graphitization process, fundamentally dictating the fundamental trend of graphitization. External conditions like temperature, pressure, and time either expedite or retard the evolution of organic matter by regulating the development of the coal structure and composition [6–8]. Owing to the influence of the metallogenic environment, the graphitization degree of natural coal-based graphite minerals exhibits significant disparities across different regions [1]. There are numerous hurdles in attempting to reverse the graphitization process using natural coal-based graphite. As a result, many scholars have resorted to laboratory simulation methods. They utilize anthracite as the precursor and simplify the pressure and time factors through the high-temperature calcination approach [9–11] to conduct forward simulations of the graphitization process and have achieved a string of outcomes. Franklin was the pioneer to conduct a high-temperature simulation experiment centered around graphitization, using anthracite as the research subject. At 2000 °C, it was ascertained that the graphite structure gradually emerged, thus verifying that temperature is an indispensable primary factor in coal graphitization. Based on the presence or absence of graphite components at 3000 °C, graphite carbon and non-graphitizable carbon were differentiated [12]. González et al. performed high-temperature simulation experiments on anthracite and established that the graphitization of anthracite is related to the molecular structure arrangement. The graphite microcrystalline particles in coal grew rapidly and their orientation was markedly enhanced between 2000 °C and 2400 °C [13]. Nyathi discovered that certain anthracites failed to develop a graphite structure even under an ultra-high temperature of 3000 °C and posited that differences exist in the macromolecular structures of coals formed in diverse coal-forming environments, which consequently result in varying graphitization capabilities among anthracites [14]. Zhang et al. heated a series of coal samples ranging from low-rank bituminous coal to anthracite and found that the carbonization and pre-graphitization stages occur between room temperature and 2000 °C, chiefly manifested as the elimination of heteroatoms and the combination of aromatic structures. Between 2000 °C and 3000 °C lies the graphitization stage, during which the three-dimensional graphite structure progressively develops and refines [15]. However, with the continuous deepening of research, the limitations of the high-temperature calcination method have gradually come to the fore. Constrained by the instrument, the maximum temperature of most experiments cannot exceed 3000 °C and cannot be sustained for an extended period. Therefore, previous studies predominantly centered on the graphitization of coal during the heating process and indirectly regarded the coal-based graphite sample obtained at the highest temperature as the endpoint of evolution, thereby streamlining the graphitization process. Nevertheless, research on whether the graphitization degree can be further enhanced after the temperature reaches 3000 °C and the subsequent graphitization process remains insufficient.

In light of the aforementioned issues, this study commences relevant experiments by taking the subsequent graphitization process of coal in high-temperature simulation as the starting point. Due to the limitations of the instrument, the maximum temperature of the experiment likewise cannot exceed 3000 °C. Consequently, through methods such as enhancing the initial metamorphic degree of coal samples and introducing a catalyst to make up for the temperature conditions [16, 17], coal samples are enabled to attain a higher graphitization degree within the energy system at a fixed temperature, thus unveiling the further evolutionary process of coal graphitization under high-temperature circumstances.

## 2. Experimental samples and methods

### 2.1. Selection and treatment

The high-temperature graphitization process and graphitization degree of coal samples with different metamorphic degrees vary significantly. Rodrigues et al. carried out continuous heating experiments on anthracites whose maximum vitrinite reflectance was 2.62%, 5.23%, and 6.25%, respectively. It was found that under the condition of 2500 °C, a three-dimensional ordered structure emerged in the high-metamorphic anthracite, while the low-metamorphic anthracite was still mainly in a disordered turbostratic structure [16]. For this reason, in this study, the high-metamorphic anthracite from the Huyan Mountain area in Shanxi was collected as the precursor, with $R_{o,max}$/% being 6.32. By increasing the initial metamorphic degree of the coal sample, it is enabled to enter the graphitization stage earlier during the heating process. The results of its proximate analysis (GB/T 212–2008) and ultimate analysis (GB/T 476–2001) are presented in Table 1.

In the process of researching coal graphitization, scholars have gradually come to understand that some minerals in coal possess the ability to influence the graphitization process. Elements like sulfur, iron, and calcium, as well as their compounds, can lower the conditions required for coal graphitization and speed up the graphitization process [18–21]. Tang et al. discovered through high-temperature simulation experiments on Taixi anthracite that using $Fe_2O_3$ as a catalyst can notably reduce the starting temperature of graphitization, accelerate the construction speed of the graphite lattice, and enhance the graphitization degree. The research asserts that the fundamental mechanism entails the incessant formation and scission of Fe-C and C-C bonds under high-temperature conditions. In view of the robust binding interplay between iron and carbon, regular aromatic six-membered rings are eventually engendered. Moreover, the dissolution-precipitation mechanism is also deemed to be of pivotal significance in promoting the graphitization process [22]. In this experiment, with reference to previous research, 200-mesh $Fe_2O_3$ powder produced by AVIC Zhongmai Metal Materials Co., Ltd. was chosen as the catalyst, and the coal sample and $Fe_2O_3$ powder were proportioned at a ratio of 8:2 [22] in order to achieve the optimal catalytic effect.

To prevent the complex mineral components in raw coal from affecting the experiment, the raw coal was subjected to acid washing and demineralization to remove the minerals contained in the coal and reduce the interference factors in the experiment [23].

### 2.2. High-temperature simulation experiment

The high-temperature equipment used in this study is the NTG-SML-60W integrated laboratory graphitization furnace, and the manufacturer is Zhuzhou Nuotian Electrothermal Technology Co., Ltd.

The experiment adopts a segmented heating method. Before heating, the gas is evacuated and replaced once at a vacuum degree of 5 Pa, and argon gas with a flow rate of 10 L/min is passed through the whole experiment for protection.

Table 1. Proximate analysis and ultimate analysis of anthracite from Huyan Mountain.

| Sample collection site | $R_{o,max}$/% | Proximate analysis | | | | Ultimate analysis | | |
|---|---|---|---|---|---|---|---|---|
| | | $M_{ad}$ (%) | $A_d$ (%) | $V_{daf}$ (%) | $FC_d$ (%) | $C_{daf}$ (%) | $H_{daf}$ (%) | H/C |
| Huyan Mountain | 6.32 | 1.06 | 13.32 | 4.69 | 82.62 | 93.79 | 0.76 | 0.097 |

In the initial stage, the temperature is increased to 1000 °C at a heating rate of 5 °C/min and kept for 60 min. Then, the temperature is raised to the target temperature point at a rate of 10 °C/min and kept for 120 min. Finally, it is naturally cooled to room temperature [24]. The set temperature range is 2100–3000 °C, with an interval of 300 °C, and a total of 4 temperature points are set.

This experiment sets up two groups of a total of 8 samples, including a group of demineralized coal samples without additive and a group of coal samples with added $Fe_2O_3$ catalyst, to conduct high-temperature simulation experiments and study the further graphitization evolution process of anthracite from Huyan Mountain at high temperatures. The specific experimental scheme is shown in Table 2.

## 2.3. X-ray diffraction analysis (XRD)

SmartLab-9kW was used as the XRD test instrument. Copper targets with a 45 kV accelerating voltage and a 200 mA current were chosen. The scan range was set as 2θ from 5° to 70°, with a scan rate of 2°/min and an X-ray wavelength of 0.15418 nm. Two diffraction peaks (2θ range 20°~30° and 40°~50°, respectively) on the XRD pattern matched the positions of the 002 and 100 peaks in the standard graphite XRD diffraction pattern. The lattice parameters (carbon layer spacing $d_{002}$, extension degree $La$, and stacking degree $Lc$) were calculated based on Bragg's equation and Scherre's formula using Jade software [25, 26].

$$d_{002} = \lambda/2sin\theta_{002} \tag{1}$$

In the formula, $d_{002}$ is the average interlayer spacing of crystallites; $\lambda$ is the wavelength of the X-ray, $\lambda = 0.154056$ nm; $\theta_{002}$ is the diffraction angle corresponding to the 002 peak, in degrees of units.

$$Lc = 1.05\lambda/\beta_{002}cos\theta_{002} \tag{2}$$

In the formula, $Lc$ is the average height of crystallites in the c-axis direction; $\beta_{002}$ is the half-width of the 002 peak.

$$La = 1.84\lambda/\beta_{100}cos\theta_{100} \tag{3}$$

In the formula, $La$ is the average diameter of the crystallites; $\beta_{100}$ is the half-width of the 100 peak, and $\theta_{100}$ is the diffraction angle corresponding to the 100 peak position, in degrees of units.

## 2.4. Laser Raman spectroscopy (Raman)

Laser Raman spectroscopy is often used to analyze organic matter [27]. A Jobin–Yvon Labram HR Evolution model high-resolution micro-Raman spectrometer was used in the Raman spectroscopic experiment. A Nd:YAG (532 nm) laser was used as the excitation light source in the experiment, with a laser power of 100 MW, a scanning range of 800–3500 cm$^{-1}$, and an exposure time of 10 s. The fitting and processing analyses were conducted with the obtained Raman spectra using the Lorentz function in the Origin8.0 software.

**Table 2. High-temperature simulation experimental scheme.**

| Temperature/°C | Number | |
| --- | --- | --- |
| | Additive-free samples | Fe₂O₃-added samples. |
| 2100 | HR-1 | HC-1 |
| 2400 | HR-2 | HC-2 |
| 2700 | HR-3 | HC-3 |
| 3000 | HR-4 | HC-4 |

The measured Raman were found in two regions, which were the first-order Raman (700–2000 cm$^{-1}$) and the second-order Raman (2000–3000 cm$^{-1}$). The primary Raman could be divided into four types of defect peaks (D1~D4 peaks) and one ordered graphite peak (G peak). The secondary Raman spectrum contained only two peaks (S1 and S2) at the low evolutionary stage, and as the evolution increased, the S1 peak gradually split into two peaks and the S2 peak disappeared [28].

Defect peaks caused by lattice defects and disordered carbon structures (active structures) within the graphite layers are called D peaks [29]. According to different causes and positions, it can be divided into 1350 cm$^{-1}$ (D1), 1620 cm$^{-1}$ (D2), 1500 cm$^{-1}$ (D3), and 1200 cm$^{-1}$ (D4), while the D3 and D4 peaks only appear in samples with high disorder [30]. The degree of development of second-order Raman is positively correlated with the degree of structural order of the three-dimensional graphite lattice [16].

In this study, parameters $R_2$ and $R_3$ are employed to characterize the degree of structural defects in the samples. These two parameters are frequently used to quantitatively assess the degree of internal lattice defects or the order degree of carbon materials [31,32].

$R_2$ is termed the "in-plane defect parameter", which is utilized to represent the proportion of in-plane defects (D1) in the graphite sheet plane. It is applicable for evaluating samples with a high degree of graphitization that have fewer defects or are mainly dominated by D1-type defect peaks. Its calculation formula is as follows:

$$R_2 = A_{D1}/A_{(G+D1+D2)} \tag{4}$$

In the formula, A is the corresponding peak area.

$R_3$ is called the "total defect parameter", which is used to characterize the percentage of all types of defects. It is suitable for evaluating samples with a low degree of graphitization that have more defects. Its calculation formula is as follows:

$$R_3 = A_{D1+D2+D3+D4}/A_{(D1+D2+D3+D4+G)} \tag{5}$$

## 2.5. High-resolution transmission electron microscopy (HRTEM)

A Tecnai G2 F30 field emission transmission electron microscope host was used in the high-resolution transmission electron microscope (HRTEM) experiment with an accelerating voltage of 300 kV, a point resolution of 0.20 nm, a line resolution of 0.10 nm, and a 0.14 nm information resolution, with a 3000- to 500000-fold magnification. The HRTEM method can be employed to characterize the graphite lattice construction process and the three-dimensional structure development process [33].

## 2.6. Scanning electron microscopy (SEM)

Scanning electron microscopy (SEM) was conducted on a ZEISS GeminiSEM 300 scanning electron microscope. The point resolution is 1.0 nm, and the range of accelerating voltage is 0.5 ~ 30 kV. The sample surfaces were pretreated by sputter-coating with gold.

## 3. Results and discussion

### 3.1. Microcrystalline structure evolution process

The samples after high-temperature heat treatment were subjected to XRD testing. The XRD patterns were fitted and processed by Jade software (Fig 1(a)). The relevant lattice parameters (carbon layer spacing $d_{002}$, extension degree $La$, and stacking degree $Lc$) were calculated using the Bragg equation and Scherre formula (Table 3).

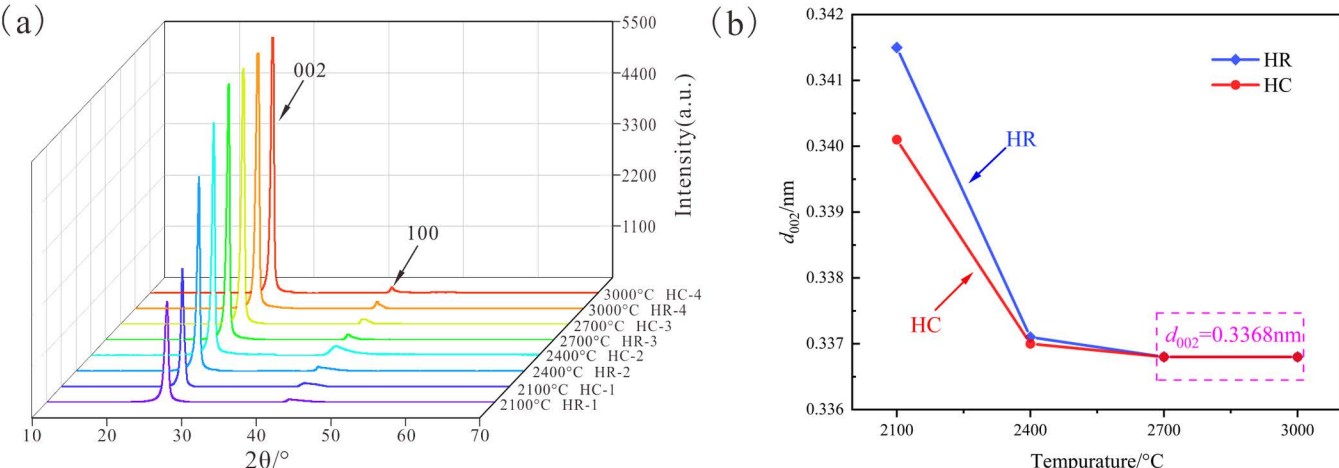

**Fig 1. XRD test results.** (a) XRD diffraction pattern; (b) Evolution curve of $d_{002}$. With the elevation of temperature, the (002) peak demonstrates a trend of approaching the peak position of the standard graphite peak (2θ = 26.6°). Concurrently, its peak intensity enhances, and the full width at half maximum ($FWHM_{002}$) diminishes. Notably, the HC sample exhibits a more acutely defined peak shape. When the temperature reaches 2700 °C, the rate of graphitization decelerates conspicuously. The (002) peak arrests its development at 2θ = 26.44°, with no discernible changes thereafter. Utilizing the Bragg equation to compute the interlayer spacing $d_{002}$ of the carbon layers, it is revealed that within the temperature range of 2100–2400 °C, as the temperature ascends, the $d_{002}$ value declines rapidly, approaching the interlayer spacing of standard graphite, which is 0.3354 nm. The HC sample possesses a lower $d_{002}$ value, indicative of a higher degree of graphitization. However, as the temperature nears 2700 °C, the rate of $d_{002}$ development gradually attenuates until it ceases. Ultimately, both the HR and HC samples reach an identical $d_{002}$ value of 0.3368 nm.

**Table 3. XRD parameter analysis result table.**

| Temperature/°C | Number | $2\theta_{002}/°$ | $FWHM_{002}/°$ | $2\theta_{100}/°$ | $Lc$/nm | $La$/nm | $d_{002}$/nm |
|---|---|---|---|---|---|---|---|
| 2100 | HC-1 | 26.18 | 0.49 | 42.54 | 16.87 | 17.38 | 0.3401 |
| | HR-1 | 26.07 | 0.53 | 42.43 | 15.58 | 17.02 | 0.3415 |
| 2400 | HC-2 | 26.43 | 0.46 | 42.37 | 17.83 | 20.61 | 0.3370 |
| | HR-2 | 26.42 | 0.50 | 42.42 | 16.43 | 20.13 | 0.3371 |
| 2700 | HC-3 | 26.44 | 0.45 | 42.43 | 18.19 | 28.83 | 0.3368 |
| | HR-3 | 26.44 | 0.48 | 42.38 | 16.91 | 28.78 | 0.3368 |
| 3000 | HC-4 | 26.44 | 0.45 | 42.39 | 18.23 | 31.31 | 0.3368 |
| | HR-4 | 26.44 | 0.46 | 42.38 | 17.91 | 30.28 | 0.3368 |

Through the analysis of XRD patterns and related lattice parameters, it can be discerned that the development process of graphite peaks in the simulated experimental samples can be partitioned into two distinct stages.

In the temperature range from 2100 °C to 2400 °C, as the thermal simulation temperature escalates, the (002) peak within the XRD pattern of the anthracite from Huyan Mountain promptly converges towards the position of the standard graphite peak (2θ = 26.6°). The peak shape becomes more pointed, the interlayer spacing $d_{002}$ of carbon layers decreases significantly, and the degree of graphitization is considerably augmented. By employing $d_{002}$ as the criterion for appraising the degree of graphitization, the $Fe_2O_3$ catalyst exhibits a pronounced promoting effect on graphitization during this interval. The (002) peak of the HC sample is closer to the standard graphite peak position in comparison to that of the HR sample, accompanied by lower $d_{002}$ and $FWHM_{002}$ values.

When the temperature surpasses 2700 °C, the position of the (002) peak of the anthracite from Huyan Mountain stabilizes at 2θ = 26.44°, with the $d_{002}$ value descending to 0.3368 nm and ceasing to decline any further, the degree of graphitization remains unaltered. However, as the temperature ascends, the $FWHM_{002}$ value of the HR sample persists in decreasing, signifying that its internal aromatic structure continues to develop and inclines towards a more orderly state [26]. The $Fe_2O_3$ catalyst exerts no conspicuous impact on the development of the carbon layer interlayer spacing in this stage, and the $d_{002}$ values of both the HC and HR samples are 0.3368 nm.

By comparing the $d_{002}$ values of the simulated samples (Fig 1(b)), it can be detected that there exists a graphitization threshold for the anthracite from Huyan Mountain, precisely $d_{002}$ = 0.3368 nm. The closer it approaches this value, the more sluggish the development rate of $d_{002}$ until it comes to a halt. Although the $Fe_2O_3$ catalyst demonstrates a remarkable pro-moting effect on the graphitization process at 2100 °C, leading to a substantially higher degree of graphitization of the HC sample than that of the HR sample and advancing the graphitization stage, with the elevation of temperature, the disparity in graphitization between the two continuously diminishes. Eventually, their degrees of graphitization become identical, and they attain the same $d_{002}$ value at 2700 °C. The introduction of the catalyst and subsequent temperature increments are incapable of reducing the $d_{002}$ value of the anthracite from Huyan Mountain below this graphitization threshold.

The variation trends of the microcrystalline parameters La and Lc of the samples with temperature are plotted (Fig 2) to dissect the development process of the high-temperature microcrystalline structure of the anthracite from Huyan Mountain.

With respect to the analysis of the stacking degree Lc of aromatic lamellae (Fig 2(a)), the HR sample exhibits a rel-atively rapid Lc growth rate throughout the entire heating process. Even after the $d_{002}$ value reaches the graphitization threshold and ceases to decrease at 2700 °C, its internal aromatic lamellae continue to stack and develop at a brisk pace. For the HC sample, its Lc value in the 2100–2400 °C range is already proximate to that of the HR sample in the 2700–3000 °C range. The catalytic promotion effect enables the Lc value to progress in advance, thereby advancing the stage of aromatic lamellae overlapping development. With the augmentation of temperature, the degree of graphitization of the HC sample nears the threshold, and its Lc value exhibits an evolutionary pattern analogous to that of the $d_{002}$ value, with the development rate gradually decelerating and approaching stagnation between 2700 °C and 3000 °C.

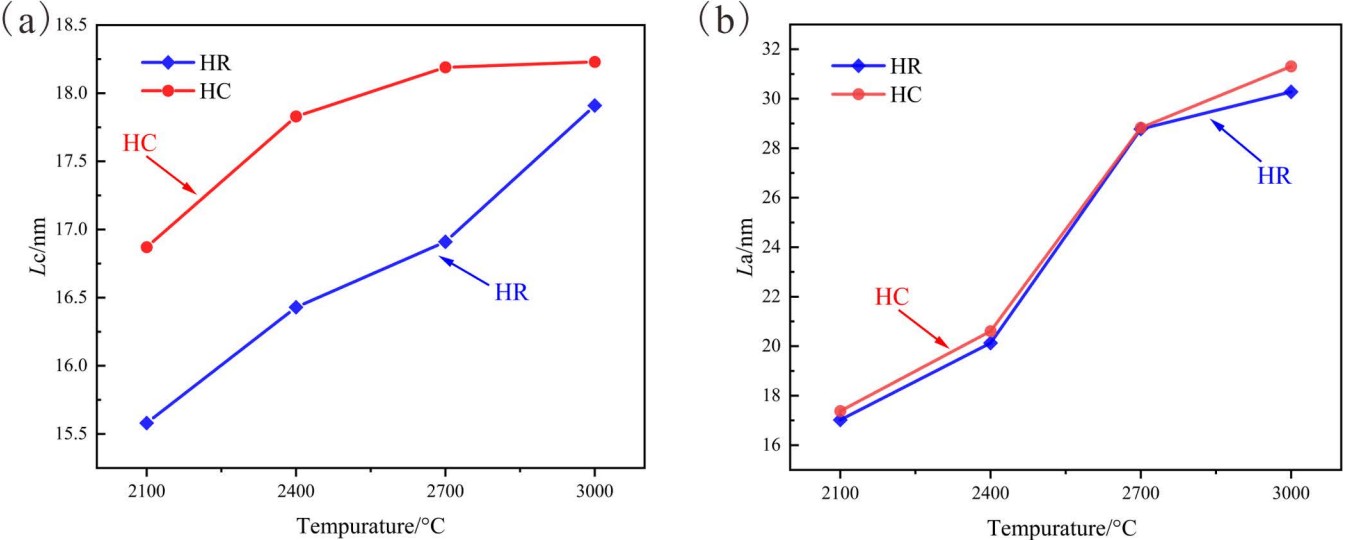

**Fig 2. Curves of microcrystalline parameters vary with temperature.** (a) Evolution curve of Lc; (b) Evolution curve of La.

In terms of the analysis of the extension degree $La$ of aromatic lamellae (Fig 2(b)), the $La$ value of the samples experiences a rapid increment between 2100 °C and 2700 °C. When the temperature attains 2700 °C, the $d_{002}$ value of the samples reaches the graphitization threshold, the growth rate of $La$ is substantially reduced, and the splicing development process of graphite microcrystals is conspicuously decelerated. Under the influence of the promoting graphitization effect of the catalyst, the $La$ value of the HC sample is marginally higher than that of the HR sample under identical conditions at each temperature. Nevertheless, the catalyst does not modify the original development trend of the graphite microcrystal layers of the anthracite from Huyan Mountain, and the HC and HR samples possess similar $La$ evolution laws.

Based on a comprehensive analysis of relevant graphite microcrystal parameters, it can be concluded that the $Fe_2O_3$ catalyst has a conspicuous promoting effect on the graphitization process of the anthracite from Huyan Mountain. It can curtail the temperature prerequisite for graphitization, permitting the anthracite to achieve a lower carbon layer interlayer spacing at the same temperature, thereby facilitating the development of graphite carbon layers. It can also accelerate the vertical overlapping of graphite microcrystals, leading to an advanced development stage and consequently enhancing the stacking degree of the graphite microcrystals of the coal. However, for the development of the size of graphite aromatic lamellae, it merely has a feeble promoting effect and does not alter the original lateral development law of the aromatic lamellae of the anthracite.

Based on the above analysis, there is a graphitization threshold for the anthracite from Huyan Mountain during the high-temperature thermal simulation process: $d_{002}$ = 0.3368 nm. When the temperature ascends above 2700 °C, the $d_{002}$ value reaches this threshold and ceases to decline, and the degree of graphitization no longer continues to escalate. Even by introducing the $Fe_2O_3$ catalyst, which has a significant promoting effect on graphitization during the initial heating stage, it is unable to modify the original graphitization trend and make the degree of graphitization of the sample surpass this threshold. Moreover, the development of the internal graphite microcrystal structure is also notably affected. The development of the extension degree $La$ and stacking degree $Lc$ of aromatic lamellae will experience a significant slowdown after reaching the graphitization threshold.

## 3.2. Graphite lattice construction process

The Raman spectra of the samples after thermal simulation treatment were obtained through fitting (Fig 3). The relevant structural characterization parameters, including the full width at half maximum of the graphite peak G ($FWHM_G$), the proportion of the in-plane defect peak $R_2$, and the proportion of the total defect peak $R_3$ were calculated (Table 4).

As the temperature in the thermal simulation process ascends, the intensity of the G peak, which symbolizes the ordered graphite structure within the sample, augments. Meanwhile, the position of this peak migrates closer to 1580 cm⁻¹. Conversely, the intensity of the D peak, which represents the disordered structure inside the sample, attenuates. This mirrors the process of defect repair within the sample and the continuous progression of the graphite lattice. The second-order Raman peak S2 gradually vanishes, signifying the establishment and enhancement of the three-dimensional graphite structure. At 2100 °C, the samples have already attained a relatively high level of graphitization. The Raman defect peaks predominantly consist of the D1-type peak, denoting in-plane defects, and the D2-type peak, signifying interlayer defects. Notably, no D3- or D4-type peaks that reflect the internal disordered structure are discernible.

The curves depicting the variations in the proportion of in-plane defects ($R_2$) and the proportion of total defects ($R_3$) of the anthracite samples from Huyan Mountain with regard to temperature are plotted (Fig 4). This is carried out to dissect the development of the internal graphite lattice and the defect repair situation of the samples.

Between 2100 °C and 2400 °C, the evolutionary tendency of $R_2$ resembles that of $d_{002}$. At 2100 °C, due to the catalytic influence, the HC sample exhibits a higher degree of graphitization and a lower proportion of layer plane defects. However, as the temperature elevates to 2400 °C, the degree of graphitization of the HC sample is progressively overtaken by the HR sample. Consequently, the disparity in graphitization between the two diminishes, and the difference in $R_2$ also

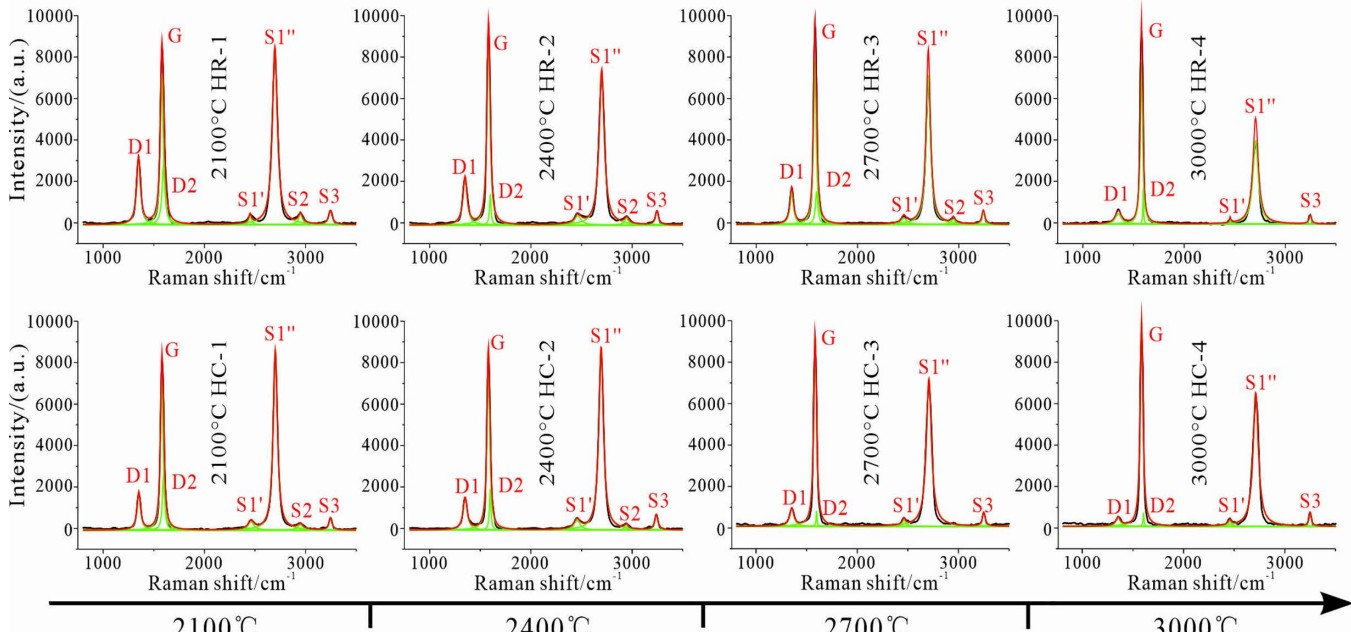

**Fig 3. Peak-fitting diagram of Raman spectra.** Concomitant with the elevation in the temperature, the intensity of the G peak (graphite peak) escalates, and its peak position drifts towards 1580 $cm^{-1}$. The intensity of the D peak (defect peak) dwindles, where in-plane defects (D1) and interlayer defects (D2) are preponderant. The second-order Raman peak, the S2 peak, vanishes completely, and the three - dimensional structure of graphite is subject to a process of incremental improvement.

**Table 4. Raman parameter analysis result table.**

| Temperature/°C | Number | $FWHM_G$ | $R_2$ | $R_3$ |
|---|---|---|---|---|
| 2100 | HC-1 | 32.65 | 0.20 | 0.37 |
| | HR-1 | 37.10 | 0.29 | 0.38 |
| 2400 | HC-2 | 33.87 | 0.19 | 0.24 |
| | HR-2 | 34.89 | 0.21 | 0.28 |
| 2700 | HC-3 | 30.93 | 0.12 | 0.14 |
| | HR-3 | 30.31 | 0.18 | 0.25 |
| 3000 | HC-4 | 28.51 | 0.07 | 0.09 |
| | HR-4 | 26.07 | 0.14 | 0.20 |

declines correspondingly. When the temperature exceeds 2400 °C, as the samples approach the graphitization threshold, the development rate of the $d_{002}$ value plummets significantly until it halts. The decreasing rate of $R_2$ of the HR sample is also affected and shows a conspicuous reduction. The correlation between $R_2$ and $d_{002}$ of the HR sample exhibits a similar developmental pattern as the temperature rises, indicating that the development of its internal microcrystalline structure and the elimination of in-plane defects occur in tandem [24]. Under the catalytic effect, the HC sample manifests a faster healing rate of lattice plane defects at temperatures above 2400 °C, and the difference in $R_2$ between it and the HR sample widens as the temperature escalates (Fig 4(a)).

When analyzing the proportion of total defects $R_3$, it can be ascertained that the $R_3$ and $R_2$ curves of the HR sample possess similar evolutionary trends. This reflects that during the high-temperature graphitization process of the anthracite

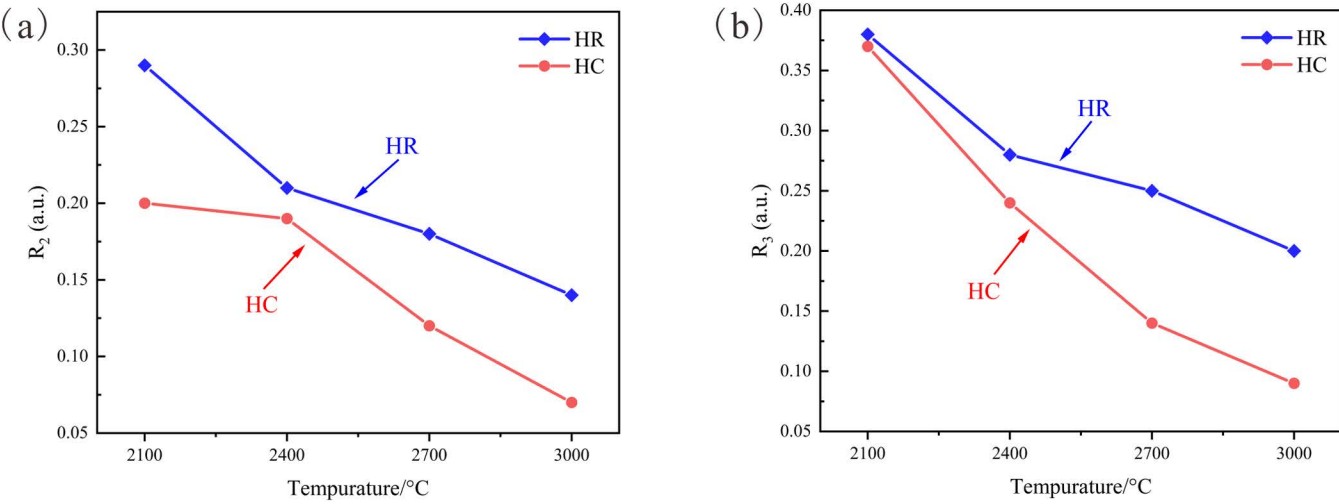

**Fig 4. Curves of Lattice parameters vary with temperature.** (a) Evolution curve of $R_2$; (b) Evolution curve of $R_3$.

from Huyan Mountain, the D1 peak is the principal defect peak, and the healing of in-plane defects is the preponderant process in the lattice development process. The $R_3$ evolution curve of the HC sample also reflects this phenomenon. It has a similar evolutionary trend with the $R_2$ curve in the 2400–3000 °C range, and the difference in $R_3$ between the samples widens as the temperature ascends. Additionally, based on the $R_3$ value of the HC sample at 2100 °C, it can be observed that the $R_2$ value of the HR sample is substantially smaller than that of the HC sample under this condition, yet their $R_3$ values are almost identical. This verifies that the $Fe_2O_3$ catalyst chiefly promotes the construction of the graphite lattice by expediting the healing of in-plane lattice defects (Fig 4(b)).

Based on a comprehensive analysis of the $d_{002}$ parameter and the Raman lattice parameters, it can be detected that after reaching the graphitization threshold, as the interlayer spacing of carbon layers ceases to decrease, the development rate of the internal lattice of the anthracite from Huyan Mountain decelerates remarkably. Nevertheless, it still evolves towards complete defect repair and the formation of a perfect graphite lattice. Moreover, the $Fe_2O_3$ catalyst empowers the samples to maintain a relatively swift pace in perfecting internal defects even after reaching the graphitization threshold by accelerating the process of in-plane defect repair, thereby effectively facilitating the development of the graphite lattice. Nevertheless, once the temperature elevates to 2700 °C, as Fe reaches its gasification temperature, a fraction of the active catalytic components emanates in a gaseous state. Subject to the combined constraints of this phenomenon and the graphitization threshold, the catalytic efficacy of $Fe_2O_3$ exhibits a perceptible decline.

### 3.3. Microstructural characteristics

High-resolution transmission electron microscopy and scanning electron microscopy was carried out on the samples after high-temperature simulation. Representative microscopic images were carefully selected and photographed. Subsequently, in-depth analysis of the microscopic morphology of graphite and the evolution process of crystal lattice was carried out (Figs 5–7).

Oberlin [34] made the discovery that the graphitization process consists of several key stages: stacking, alignment, wrinkling removal, and extension. Zheng [35] classified the graphitization process of coal into four specific stages, namely the aromatic layer graphite stage, micro-column graphite stage, wrinkled graphite stage, and flat graphite stage. It is noted that as the degree of metamorphism increases, the basic structure unit (abbreviated as BSU) of coal gradually evolves from the vortex-like structure typical of anthracite into a flat graphite layer.

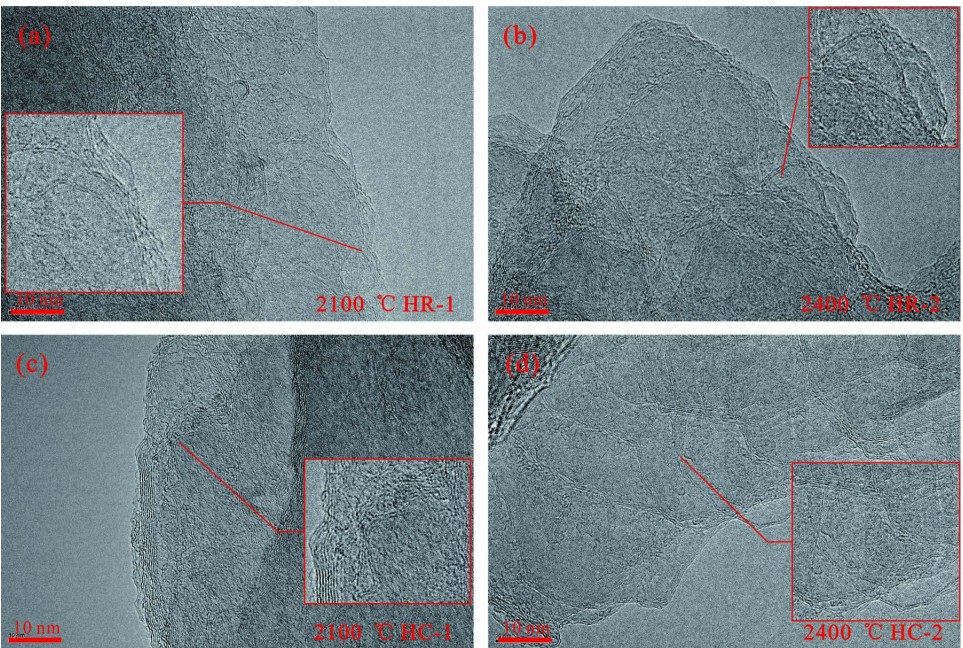

**Fig 5. HRTEM characterization between 2100 °C and 2400 °C.** (a) HR-1 at 2100 °C; (b) HR-2 at 2400 °C; (c) HC-1 at 2100 °C; (d) HC-2 at 2400 °C. Wrinkled graphite stage, the number of stacked carbon layers is relatively meager. The graphite laminae present an abundance of imperfections. A substantial quantity of disordered carbon layers is discernible.

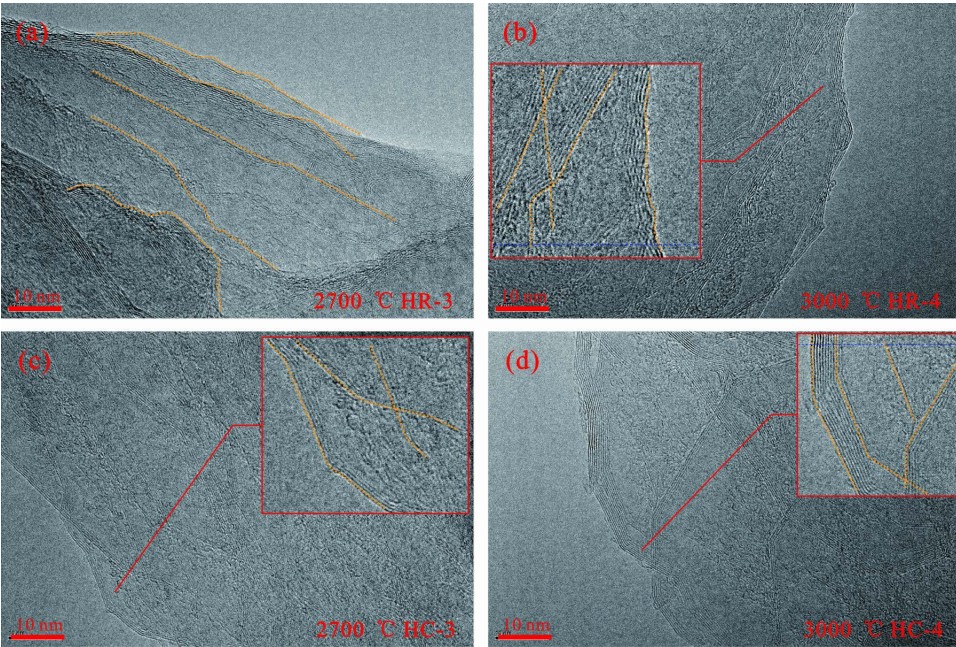

**Fig 6. HRTEM Characterization between 2700 °C and 3000 °C.** (a) HR-3 at 2700 °C. In the transitional regime from wrinkled graphite to flat graphite, the malleability of the graphite laminae augments. The external orientation becomes markedly pronounced, while the internal configuration remains tortuous and chaotic; (b) HR-4 at 3000 °C. In the transitional regime from wrinkled graphite to flat graphite, in comparison to the state at 2700 °C, the graphite lamellae experience further evolution. The orientation becomes more pronounced with the number of stacked carbon layers escalates, and the quantity of disordered carbon layers dwindles; (c) HC-3 at 2700 °C. Flat graphite stage, the edges of the graphite lamellae manifest a remarkable degree

of regularity. The orientation is highly conspicuous, and the quantity of stacked layers is relatively scant, spanning approximately 3 to 5 layers. Intrinsically, disordered carbon layers endure; (d) HC-4 at 3000 °C. Flat graphite stage, the structural order experiences a further augmentation. The amount of disordered carbon layers dwindles substantially, and the number of stacked layers ascends to upwards of ten layers.

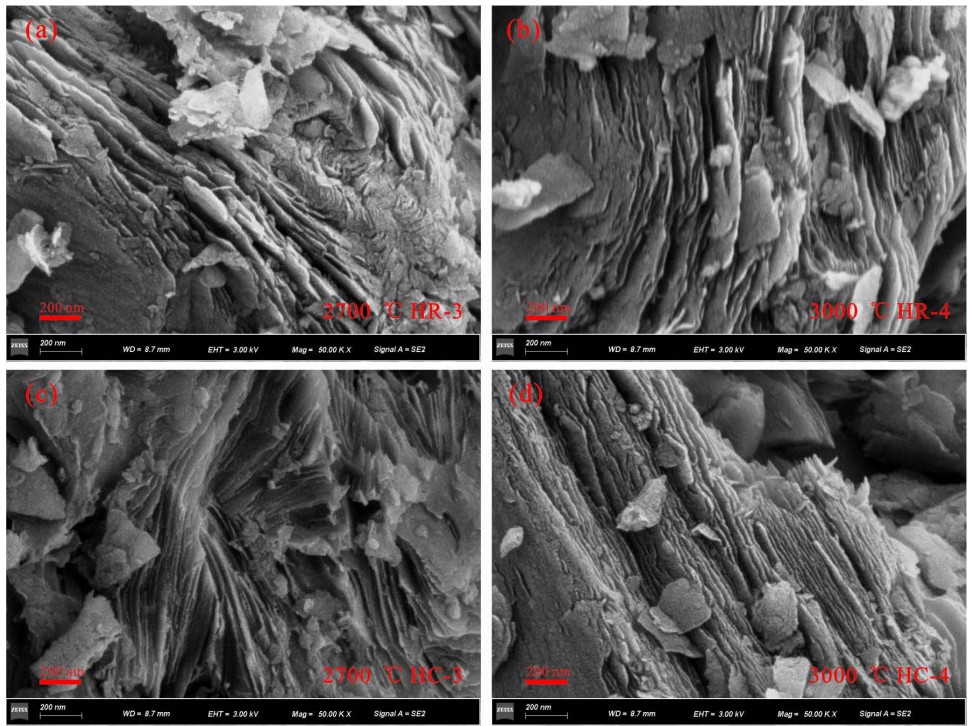

**Fig 7. SEM characterization between 2700 °C and 3000 °C.** (a) HR-3 at 2700 °C; (b) HR-4 at 3000 °C; (c) HC-3 at 2700 °C; (d) HC-4 at 3000 °C.

During the heating process of the anthracite from Huyan Mountain, the main activities involve the defect elimination of graphite lamellae and the optimization of the lattice. In the temperature range of 2100 °C to 2400 °C, both the HC and HR samples are found to be in the wrinkled graphite stage. At this stage, although a graphite structure has been formed, the edges of the graphite lamellae are curved and not well-defined. The number of stacked carbon layers is rather limited, and a significant number of disordered carbon layers can be clearly observed (Fig 5).

As the thermal simulation temperature escalates, under microscopic observation, the process of the gradual formation and perfection of graphite flakes can be discerned, accompanied by a successive augmentation in the length of carbon layers and the thickness of stacked layers.

Both the HR sample and the HC sample have succeeded in generating relatively consummate three-dimensional graphite structures at 2700 °C, as illustrated in Fig 6(a) and (c). Notwithstanding the attainment of the graphitization threshold at this temperature, the internal graphite structures of the samples persist in evolving with subsequent temperature hikes.

In the instance of the HR sample, at 2700 °C, the graphite flakes that have been formed possess a relatively scanty number of stacked carbon layers. The orientation of the flakes becomes more conspicuous at the edges, yet erratic bends are observable in some regions. The closer one approaches the interior of the graphite flakes, the more indistinct the orientation is, and the edges display curvature (Fig 6(a)). When the temperature reaches 3000 °C, the number of stacked

carbon layers in the graphite flakes multiplies, attaining a maximum of over ten layers. The edges become relatively straight, exhibiting a lucid orientation. Both the development and stacking of the internal graphite flakes have progressed, and the internal defects have lessened (Fig 6(b)).

For the HC sample, due to the catalytic impetus, the development and stacking of its graphite flakes surpass those of the HR sample under equivalent conditions. At 2700 °C, the sample has a relatively sparse number of stacked layers, approximately 3–5 layers, and copious disordered carbon layers endure within (Fig 6(c)). When the temperature ascends to 3000 °C, the number of graphite stacked layers proliferates to over ten. The quantity of disordered carbon layers within the flakes dwindles, and the edges become more rectilinear with a more pronounced orientation. A distinct and orderly stacking of graphite flakes can be perceived. The orderliness and development of its graphite structure have been augmented with the temperature increment (Fig 6(d)).

Throughout the entire temperature spectrum, the HC sample displays a more preferable formation of graphite flakes in contrast to the HR sample at the identical temperature. It exhibits a greater degree of orderliness, featuring a reduced curvature of the internal carbon layers and more straightened edge carbon layers. At 3000 °C, neither the HC nor the HR sample manifests a large-scale thick-layer graphite stacking architecture. Instead, they predominantly exhibit small-scale thin-layer graphite stacking configurations, with a maximum of approximately ten layers and typically three to five layers.

The morphological development of graphite crystallites observed by SEM correlates well with HRTEM analyses. Although the $d_{002}$ parameter stops further decreasing at 2700 °C, continuous structural evolution is evident in the graphite crystallites. At this temperature, the sample forms distinct graphitic lamellar structures with nearly parallel stacking of graphite flakes. However, these flakes remain small in size and display disordered, twisted configurations (Fig 7(a), (c)). Upon increasing the temperature to 3000 °C, the graphite flakes grow larger and evolve into dense, straight lamellae. The interlayer stacking becomes more parallel, accompanied by a marked enhancement in structural ordering (Fig 7(b), (d)). Notably, the HC sample at 3000 °C demonstrates more ordered and compact stacking of graphite flakes compared to the HR sample, with improved parallelism between layers and a higher degree of crystallite development.

Via microstructural analysis, it can be discerned that subsequent to the coal sample reaching its graphitization threshold, although the $d_{002}$ value halts its decline, the internal graphite lattice structure still proceeds with its development. This is manifested by an increment in the number of stacked carbon layers within the graphite flakes, the mending of internal defects, the augmentation of structural orderliness, along the even stacking and conspicuous orientation at the edges of the flakes. Such observations are congruent with the analysis centered around the microcrystalline parameters and lattice parameters.

### 3.4. Discussion

The standard graphite model is established based on Sri Lankan graphite, which is characterized by a carbon layer spacing ($d_{002}$) of 0.3354 nm. Prior research has revealed that the closer the $d_{002}$ value is to that of standard graphite, the higher the graphitization degree of coal based graphite becomes. Concurrently, significant improvements are detected in its electrical conductivity, initial discharge specific capacity, and thermal conductivity. Notably, coal based graphite manifests a more remarkable electrochemical energy-storage capacity than natural graphite [2]. Furthermore, the coal based graphene synthesized through the Hummers redox method exhibits larger microcrystalline lamellar structures and fewer defects. These characteristics endow it with distinct advantages in relevant applications. However, the existence of the graphitization threshold impedes the enhancement of the industrial properties of coal based graphite, posing an obstacle to its development and utilization [36]. Therefore, delving into the emergence mechanism of this threshold and exploring effective ways to surmount it are of utmost significance for the advancement of practical applications of coal based graphite.

The investigation of carbonaceous material graphitization has historically recognized distinct graphitization propensities among organic precursors, leading to their classification as either graphitizable carbons (GCs) or non-graphitizable

carbons (NGCs) [12]. GCs predominantly derive from hydrogen-enriched precursors, wherein hydrogen atoms mediate the disruption of interlamellar cross-linking within anthracitic aromatic domains, thereby promoting structural reorganization into graphitic configurations [10,37]. These precursors favor the nucleation of oriented polycyclic aromatic hydrocarbon assemblies during pyrolysis. Conversely, oxygen-laden precursors impede the development of ordered polyaromatic stacking architectures due to enhanced cross-linking via oxygen-mediated covalent bonding [38, 39]. This dichotomy fundamentally originates from the macromolecular heterogeneity inherent to coal macerals: Vitrinite, characterized by elevated hydrogen content, oxygenated moieties, and aliphatic substituents, exhibits relatively low aromaticity and modest aromatic condensation indices. Under elevated thermal regimes, labile bonds at aromatic peripheries undergo cleavage, facilitating progressive aromatic condensation and enhanced molecular alignment through structural reconfiguration. Inertinite, by contrast, manifests oxygen-rich compositions with larger basal structural units (BSUs), predominance of etheric oxygen linkages, minimal peripheral reactive sites, constrained structural plasticity, and superior thermal resilience.

Petrographic analysis of Huyan Mountain anthracite reveals abundant fusinitic components retaining protogenetic cellular morphology (Fig 8), corroborated by ultralow hydrogen content (0.76 wt%), indicative of substantial inertinite predominance. The robust covalent architecture of inertinite resists thermal degradation even under extreme temperatures, thereby kinetically hindering the alignment of aromatic clusters into long-range graphitic ordering. Synthesizing extant literature, we posit that the observed graphitization threshold arises intrinsically from the recalcitrance of inertinitic components to undergo graphitization under high-temperature conditions.

Notably, comparative high-pressure, high-temperature simulations by Liu et al. demonstrated that inertinite graphitization exhibits greater influence to pressure than temperature, consistent with kinetic control mechanisms. The rupture, rotation, and realignment of inertinite's large aromatic lamellae cross-linked structures all require pressure to overcome inherent structural rigidity. Critically, once critical pressure-temperature thresholds are attained, inertinite undergoes accelerated graphitization through stress-mediated structural reorganization, achieving rapid transformation into coal-bassed graphite [40].

## 4. Conclusions

This research took anthracite from Huyan Mountain, Shanxi Province as the starting material. After being subjected to acid washing and demineralization procedures, high-temperature thermal simulation experiments were carried out in the temperature range from 2100 °C to 3000 °C to explore its graphitization process under high-temperature conditions. Moreover, a control group with the addition of $Fe_2O_3$ catalyst, which has a promoting impact on graphitization, was set up

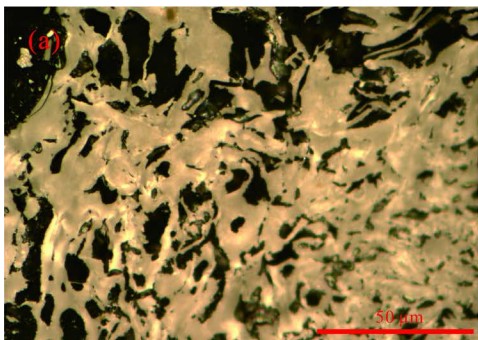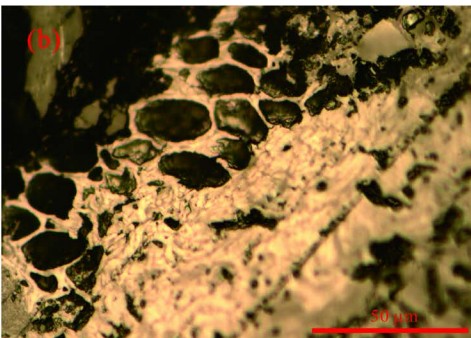

**Fig 8. Microscopic component images of anthracite from Huyan Mountain.** (a) Fusinite, oil-immersion reflected light; (b) Fusinite, cellular structure, oil-immersion reflected light.

to probe into its further graphitization capacity and process. The following conclusions were obtained through testing and analysis:

(1) The XRD results manifest that there is a graphitization threshold for the anthracite from Huyan Mountain, namely $d_{002}$ = 0.3368 nm. Once this threshold is attained, the interlayer spacing of carbon layers halts its progression. Furthermore, neither further increasing the temperature nor introducing a catalyst can surmount this threshold.

(2) The analysis of microcrystalline structure parameters reveals that after the $d_{002}$ value reaches 0.3368 nm and ceases to decrease, the internal aromatic structure of the Huyan Mountain anthracite continues to develop. However, the growth rates of the aromatic layer extension degree $La$ and the stacking degree $Lc$ slow down considerably.

(3) The Raman parameters imply that after reaching the graphitization threshold, the development rate of the internal lattice of the Huyan Mountain anthracite decelerates remarkably. Nevertheless, it still proceeds towards the complete repair of defects and the formation of a flawless three-dimensional graphite structure.

(4) The results of microstructural characteristics validate the analysis of microcrystalline structure and lattice development. After reaching the graphitization threshold, the internal graphite structure persists in evolving. This is exhibited by an increment in the number of stacked carbon layers within the graphite flakes, the mending of internal defects, the augmentation of structural orderliness, and the flat and orderly stacking with a conspicuous orientation at the edges of the flakes.

## Supporting information

**S1 Data.**
(ZIP)

## Author contributions

**Data curation:** Gaojian Chen, Fengchu Liao.

**Formal analysis:** Gaojian Chen, Simai Peng.

**Funding acquisition:** Daiyong Cao, Hongsheng He, Anmin Wang.

**Methodology:** Daiyong Cao, Simai Peng.

**Resources:** Daiyong Cao, Anmin Wang.

**Supervision:** Daiyong Cao, Hongsheng He.

**Writing – original draft:** Gaojian Chen.

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
