## [Decision Letter · Decision Letter 0]

10 Dec 2024

PONE-D-24-53055Preparation and characterization of coal-based graphite from Huyan mountain anthracite by high-temperature simulationPLOS ONE

Dear Dr. Cao,

Thank you for submitting your manuscript to PLOS ONE. After careful consideration, we feel that it has merit but does not fully meet PLOS ONE’s publication criteria as it currently stands. Therefore, we invite you to submit a revised version of the manuscript that addresses the points raised during the review process.

We look forward to receiving your revised manuscript.

Kind regards,

Anil Kumar Reddy Police

Academic Editor

PLOS ONE

“This study is financially supported by the National Natural Science Foundation of China (grant No. 42072197), the Fundamental Research Funds for the Central Universities (2023ZKPYDC07) and Projects of the Key Research and Development Program in Hunan Province (grant No. 2022SK2090).”

Reviewers' comments:

Reviewer's Responses to Questions

**Comments to the Author**

1. Is the manuscript technically sound, and do the data support the conclusions?

Reviewer #1: No

Reviewer #2: Partly

Reviewer #3: Partly

2. Has the statistical analysis been performed appropriately and rigorously? 

Reviewer #1: I Don't Know

Reviewer #2: Yes

Reviewer #3: No

3. Have the authors made all data underlying the findings in their manuscript fully available?

Reviewer #1: Yes

Reviewer #2: Yes

Reviewer #3: No

4. Is the manuscript presented in an intelligible fashion and written in standard English?

Reviewer #1: Yes

Reviewer #2: Yes

Reviewer #3: No

5. Review Comments to the Author

Reviewer #1: The study shows that coal can be efficiently converted into graphite using high temperatures and catalysts, but there are limits to how much the process can be improved. These findings can help industries produce high-quality graphite for use in batteries, electronics, and other high-tech applications. The following questions should be addressed and incorporated in the MS before re-review.

1. The manuscript is well-organised and contains much informative about the technique and outcomes. Some portions, such as challenges in coal graphitization, it couldn’t explicitly state why overcoming the graphitization threshold is critical and how this research extends existing understanding.

2. Why was Fe₂O₃ specifically chosen as the catalyst in this study? Were other catalysts considered, and how does Fe₂O₃ compare in terms of efficiency or relevance to the study's objectives?

3. Why is it significant that the Fe₂O₃ catalyst facilitates faster defect repair? Explicitly linking these findings to real-world significance would improve the focus.

4. The manuscript mentions a fixed coal-to-catalyst ratio (8:2). Was this ratio optimized through preliminary experiments, or was it based on literature? How might varying this ratio affect the results?

5. It is stated that the Fe₂O₃ catalyst accelerates defect repair and stacking processes but does not alter the graphitization threshold. What is the underlying mechanism limiting further graphitization, and can other methods (e.g., increased pressure or alternative catalysts) potentially overcome this limitation?

6. Was there any observable degradation or loss of catalytic activity for Fe₂O₃ over the course of the experiment?

7. The acid washing and demineralization process is described in detail, but could the presence of residual impurities impact the graphitization threshold? Was this evaluated?

8. How does the decrease in interlayer spacing (d002) relate to potential applications in graphite-based technologies?

9. Figure1 shows the evolution curve of d002, but the caption could discuss its implications for achieving the graphitization threshold, there is a lack of clarity.

10. Could the Fe₂O₃ catalyst's ability to promote graphitization be extended to lower-quality anthracite or other forms of coal like bituminous and low-grade coal? If so, what adjustments to the methodology might be required?

Reviewer #2: The following comments must be considered for accepting the manuscript for publication.

1. Figure captions are missing.

2. XPS data needs to be added.

3. Element compositions (C, H, N, and S) of the coal-based graphite need to be mentioned.

4. FTIR and SEM need to be carried out.

Reviewer #3: (i) Include the discussion about the role of catalyst (Fe2O3) and compare with the other available catalysts and conditions reported in literature.

(ii) All the spectra given in the Figure 3 are very low resolution and hard to read or analysis.

(iii) Very low resolution images are used in other figure also. Give high resolution images (Fig 5, Fig 6)

(iv) Novelty of this work is poorly described.

(v) The manuscript needs improvement scientifically.

6. PLOS authors have the option to publish the peer review history of their article (what does this mean? ). If published, this will include your full peer review and any attached files.

**Do you want your identity to be public for this peer review?** For information about this choice, including consent withdrawal, please see our Privacy Policy .

Reviewer #1: No

Reviewer #2: **Yes: ** Aruna Kumar Barick

Reviewer #3: No

---

## [Author Response · Author response to Decision Letter 1]

4 Mar 2025

Dear Reviewers:

We would like to begin with our sincere appreciation for all the valuable comments, insightful suggestions and thoughtful corrections offered by the reviewers to our manuscript “Preparation and characterization of coal-based graphite from Huyan mountain anthracite by high-temperature simulation” (PONE-D-24-53055). The comments and suggestions definitely helped us to improve the quality of the manuscript. We have revised the manuscript in which major changes are highlighted in red in “Revised Manuscript with Track Changes”. These changes are summarized below following a point-by-point response to reviewer's comments. Some pictures cannot be uploaded to this website. You can find them in “Response to editor and reviewers”.

Reviewer #1

Comment 1: The manuscript is well-organised and contains much informative about the technique and outcomes. Some portions, such as challenges in coal graphitization, it couldn’t explicitly state why overcoming the graphitization threshold is critical and how this research extends existing understanding.

Response: The standard graphite model is established based on Sri Lankan graphite, which is characterized by a carbon layer spacing (d002) of 0.3354 nm. Prior research has revealed that the closer the d002 value is to that of standard graphite, the higher the graphitization degree of coal based graphite becomes. Concurrently, significant improvements are detected in its electrical conductivity, initial discharge specific capacity, and thermal conductivity. Notably, coal based graphite manifests a more remarkable electrochemical energy-storage capacity than natural graphite [1]. Furthermore, the coal based graphene synthesized through the Hummers redox method exhibits larger micro-crystalline lamellar structures and fewer defects. These characteristics endow it with distinct advantages in relevant applications. However, the existence of the graphitization threshold impedes the enhancement of the industrial properties of coal based graphite, posing an obstacle to its development and utilization [2]. Therefore, delving into the emergence mechanism of this threshold and exploring effective ways to surmount it are of utmost significance for the advancement of practical applications of coal based graphite. (Line 452-465)

[1]Xu X, Cao D, Wei Y, Wang A, Chen G, Wang T, et al. Impact of graphitization degree on the electrochemical and thermal properties of coal. ACS Omega. 2024;9(2):2443-2456. https://doi.org/10.1021/acsomega.3c06871

[2]Tang Y, Che Q, Li R, Ma P, Luo P, Ju M, et al. Evolutionary characteristics of coal based graphene structure in West Guizhou and Hanpo’ao,Hunan. Journal of China Coal Society. 2023;48(1):357-372. (in Chinese). https://doi.org/10.13225/j.cnki.jccs.L022.1275

Comment 2: Why was Fe2O3 specifically chosen as the catalyst in this study? Were other catalysts considered, and how does Fe2O3 compare in terms of efficiency or relevance to the study's objectives?

Response: Our research team has long been committed to exploring the influence of SiO₂ on the coal graphitization process. As a prevalent mineral in coal, SiO₂ manifests markedly different action mechanisms under high - temperature conditions and high - temperature and high - pressure environments. Leveraging these research achievements, we have published two papers [1-2], which have not only disseminated our findings but also opened up new avenues for further research. Fe₂O₃, another common mineral in natural coal deposits, has been demonstrated by studies conducted by scholars such as Yu [3] and Tang [4] to exert a significant promotional effect on graphitization under high - temperature settings. Inspired by these prior investigations, our research group embarked on a series of experimental endeavors. Specifically, we carried out high - temperature thermal simulation experiments and high - temperature and high - pressure simulation experiments on Fe₂O₃, with the intention of drawing a comprehensive comparison between the two experimental scenarios. Through the high - temperature thermal simulation experiments, we successfully identified the existence of the graphitization threshold, which constitutes the core research content of the present paper. Currently, the body of research on the impact of minerals on graphitization remains relatively limited. In particular, there are notable inadequacies in the systematic analysis of the catalytic effects of diverse minerals. In response to these research gaps, our research group has meticulously devised an experimental plan. This plan involves conducting batch high - temperature graphitization simulation experiments on a range of minerals, including S, F, TiO₂, Fe₃O₄, and Al₂O₃. The overarching objective is to comprehensively summarize the impact of various minerals on graphitization in the future, thereby contributing to the further enrichment and refinement of research in this field.

[1]Chen G, Cao D, Wang A. Wei Y, Liu Z, Zhao M. A high-temperature thermal simulation experiment for coal graphitization with the addition of SiO2. Minerals. 2022;12(9):1239. https://doi.org/10.3390/min12101239

[2]Chen G, Cao D, Wang A. Wei Y, Liu Z, Zhao M. High-temperature and high-pressure simulation of coal graphitization with SiO2 additive. Journal of Mining Science and Technology. 2024;9(2):144-155. (in Chinese) https://doi.org/10.19606/j.cnki.jmst.2024.02.002

[3]Yu Z, Xie W, Qiu T, Lu Q, Jiang H, He Y. Effect of additives on microstructure of coal-based graphite. Coal Science and Technology. 2023; 51(5):302−308. (in Chinese) https://doi.org/10.13199/j.cnki.cst.2021-0966

[4]Tang L, Mao Q, You Z, Yao Z, Zhu X, Zhong Q, et al. Catalytic graphitization in anthracite by reduced iron particles and investigating the mechanism of catalytic transformation via molecular dynamics. Carbon. 2022;188:336-348. https://doi.org/10.1016/j.carbon.2021.12.031

Comment 3: Why is it significant that the Fe2O3 catalyst facilitates faster defect repair? Explicitly linking these findings to real-world significance would improve the focus.

Response: The phenomenon where the Fe2O3 catalyst accelerates the defect - repair rate substantiates two key aspects. Firstly, the coal sample from Huyan Mountain has not genuinely attained the upper limit of graphitization and ceased its development process as indicated by the XRD data. In actuality, this data reveals the existence of lattice defects within the Huyan Mountain coal sample, which require further amelioration. This implies that it still harbors substantial development potential and room for improvement. Secondly, this phenomenon comprehensively demonstrates the catalytic effect of Fe2O3 and lucidly illustrates the specific process and trajectory of its role in promoting graphitization. This finding holds certain reference significance for the subsequent analysis of the mineralization of natural coal based graphite under the influence of minerals, as well as for the simulation research on the graphitization induced by artificial catalysts.

After careful consideration, the fact that Fe2O3 promotes the healing of in - plane defects is merely a phenomenon observed during the experiments. Additionally, the central focus of this paper lies in the graphitization threshold rather than the catalytic function. Consequently, it is inappropriate to include this as a conclusion. Therefore, the previous conclusion 5 has been expunged and replaced.

Comment 4: The manuscript mentions a fixed coal-to-catalyst ratio (8:2). Was this ratio optimized through preliminary experiments, or was it based on literature? How might varying this ratio affect the results?

Response: Drawing on the high temperature graphitization simulation experiments conducted by Tang [1], in which Fe2O3 was employed as the catalyst, a proportion of 8:2 was demonstrated to yield highly satisfactory experimental outcomes. Concerning the catalyst proportion, Yu [2], through high temperature thermal simulation experiments, revealed that an increase in the catalyst content generally led to a higher degree of development of the coal based graphite produced. Nevertheless, research by Luo [3] indicated that when the catalyst content surpassed 20%, it had the potential to impede the development of the graphite microcrystalline structure.

Notably, in all previous investigations within our research group on the impact of minerals on graphitization, a ratio of 8:2 was consistently selected, and these endeavors invariably achieved favorable results. In light of the above - mentioned factors, a ratio of 8:2 was determined to be the optimal choice for the present study.

[1]Tang L, Mao Q, You Z, Yao Z, Zhu X, Zhong Q, et al. Catalytic graphitization in anthracite by reduced iron particles and investigating the mechanism of catalytic transformation via molecular dynamics. Carbon. 2022;188:336-348. https://doi.org/10.1016/j.carbon.2021.12.031

[2]Yu Z, Xie W, Qiu T, Lu Q, Jiang H, He Y. Effect of additives on microstructure of coal-based graphite. Coal Science and Technology. 2023; 51(5):302−308. (in Chinese) https://doi.org/10.13199/j.cnki.cst.2021-0966

[3]Luo P , Tang Y , Li R ,et al.Effects of Minerals Type and Content on the Synthetic Graphitization of Coal: Insights from the Mixture of Minerals and Anthracite with Varied Rank. Minerals, 2023, 13(8). https://doi.org/10.3390/min13081024.

Comment 5: It is stated that the Fe2O3 catalyst accelerates defect repair and stacking processes but does not alter the graphitization threshold. What is the underlying mechanism limiting further graphitization, and can other methods (e.g., increased pressure or alternative catalysts) potentially overcome this limitation?

Response: Thank you for your suggestion. Based on microscopic analysis and previous studies, we have made speculations on the reasons for the appearance of the graphitization threshold in the discussion. We believe that the main reason for this phenomenon is due to the influence of the graphitization ability of the macerals in coal. �Line 466-499�Moreover, existing studies suggest that in order to break the graphitization limitations of macerals, more often than not, it requires the synergistic effect of high temperature and high pressure rather than a single temperature factor. This view is consistent with the formation of coal-measure graphite in nature. Minerals only play an auxiliary role in the graphitization process, and we believe that even with a better catalyst, the threshold will not be broken.

Comment 6: Was there any observable degradation or loss of catalytic activity for Fe2O3 over the course of the experiment?

Response: Performing detections remains an arduous task. One of the long - standing conundrums that have perpetually confounded our research group is the means of accomplishing real - time observation and recording throughout high - temperature and high - temperature - high - pressure experiments. Over the years, our research group has endeavored to adapt the existing equipment for gas collection and measurement. Nevertheless, the outcomes have fallen far short of expectations. Additionally, operations within high - temperature and high - temperature - high - pressure conditions, such as 3000 °C or 1200 °C - 2 GPa, are rife with peril.

Notwithstanding, upon scrutinizing the evolution curves of parameters including La/Lc/R2/R3 within the temperature span from 2700-3000 °C, a decrease in the graphitization rate compared with 2400-2700 °C was discerned. It is postulated that the principal causes for this decrease are as follows: ① the constraint imposed by the graphitization threshold; ② subsequent to the temperature reaching 2800°C, Fe undergoes vaporization, and a portion of it is expelled from the high - temperature system along with the gas.�Line 354-357

Comment 7: The acid washing and demineralization process is described in detail, but could the presence of residual impurities impact the graphitization threshold? Was this evaluated?

Response: In actuality, subsequent to the demineralization process, XRD testing was performed to validate the efficacy of demineralization. As depicted in Figure 1, apart from the silicon peaks (calibrate with Si), solely the characteristic peaks of carbon (highly metamorphosed anthracite) are discernible, with no conspicuous mineral impurity peaks in evidence.

Figure 1. XRD Pattern of the Demineralized Coal Sample

Comment 8: How does the decrease in interlayer spacing (d002) relate to potential applications in graphite-based technologies?

Response: Currently, within the identification index system for coal based graphite adopted in China [1], d₀₀₂ serves as a crucial parameter for gauging the degree of graphitization. The closer the value of d002 approaches 0.3354 nm, the higher the degree of graphitization is manifested. This also implies that the structural characteristics of the experimental sample bear a greater resemblance to those of standard graphite. As stated in Response 1, under such circumstances, the electrical conductivity of the sample is correspondingly more excellent. Once the d002 value attains a specific standard, this sample can be directly exploited as graphite resources or applied in relevant fields such as graphene preparation.

[1]NB/T 11441-2023, Guidelines for identificationand quality evaluation of coal-measures graphite. China

Comment 9: Figure 1 shows the evolution curve of d002, but the caption could discuss its implications for achieving the graphitization threshold, there is a lack of clarity.

Response: We sincerely appreciate your suggestions. The captions of numerous figures in the text have been augmented. Specifically, a supplementary description regarding the XRD evolution process of the sample has been incorporated into the caption of Figure 1.�Line 202-214

Comment 10: Could the Fe₂O₃ catalyst's ability to promote graphitization be extended to lower-quality anthracite or other forms of coal like bituminous and low-grade coal? If so, what adjustments to the methodology might be required?

Response: Dear esteemed expert, we are extremely pleased to share with you certain insights concerning the influence of minerals on the graphitization process. Drawing upon prior research works and the investigations carried out by our own team, within the context of high - temperature thermal simulation experiments, when additives such as Fe₂O₃, SiO₂, Ni, and TiO₂ are employed, and provided that an appropriate temperature condition (exceeding 2400 °C) is maintained over a sufficient duration, graphitization can be successfully induced to reach the coal based graphite phase. This outcome holds true regardless of whether the initial coal sample is of low - rank bituminous type or highly metamorphosed anthracite.

Conversely, in the case of using additives such as F and S, a contrary effect is observed. The degree of graphitization of the samples obtained subsequent to the heating process is found to be lower than that of samples without any additives. Evidently, these particular additives exert an inhibitory influence on the graphitization process.

Furthermore, high - temperature and high - pressure graphitization simulation experiments entail a more rigorous experimental setup. Owing to the highly enclosed and high - pressure environment characteristic of such experiments, only coal samples of a rank equal to or higher than lean coal are eligible for experimental use. This is because, during the evolutionary process of lower - ranked coal samples, the gases generated as a result of deoxidation and dehydrogenation reactions cannot be effectively vented, thereby potentially leading to hazardous situations.

In high - temperature and high - pressure experimental scenarios, certain additives, for instance, Fe₂O₃ and Ni, continue to exhibit catalytic properties. This principle equally holds true for the process of synthesizing diamond from graphite under high - temperature and high - pressure conditions, in which ferro - nickel alloy is employed as a catalyst. The underlying reason lies in the fact that both processes essentially entail the evolution of carbon - based structures. However, additives such as SiO₂ display an effect contrary to that observed in high - temperature experiments, with an inhibitory effec

---

## [Decision Letter · Decision Letter 1]

24 Mar 2025

Preparation and characterization of coal-based graphite from Huyan mountain anthracite by high-temperature simulation

PONE-D-24-53055R1

Dear Dr. Cao,

We’re pleased to inform you that your manuscript has been judged scientifically suitable for publication and will be formally accepted for publication once it meets all outstanding technical requirements.

Kind regards,

Anil Kumar Reddy Police

Academic Editor

PLOS ONE

Additional Editor Comments (optional):

Reviewers' comments:

Reviewer's Responses to Questions

**Comments to the Author**

1. If the authors have adequately addressed your comments raised in a previous round of review and you feel that this manuscript is now acceptable for publication, you may indicate that here to bypass the “Comments to the Author” section, enter your conflict of interest statement in the “Confidential to Editor” section, and submit your "Accept" recommendation.

Reviewer #2: All comments have been addressed

Reviewer #3: All comments have been addressed

2. Is the manuscript technically sound, and do the data support the conclusions?

Reviewer #2: Yes

Reviewer #3: Partly

3. Has the statistical analysis been performed appropriately and rigorously? 

Reviewer #2: N/A

Reviewer #3: I Don't Know

4. Have the authors made all data underlying the findings in their manuscript fully available?

Reviewer #2: Yes

Reviewer #3: Yes

5. Is the manuscript presented in an intelligible fashion and written in standard English?

Reviewer #2: Yes

Reviewer #3: Yes

6. Review Comments to the Author

Reviewer #2: Though all the reviewers comments and suggestions are not responded to, the manuscript may be accepted in its current form.

Reviewer #3: The authors have addressed the comments in the revised manuscript. Therefore I do not have any further comment.

7. PLOS authors have the option to publish the peer review history of their article (what does this mean? ). If published, this will include your full peer review and any attached files.

**Do you want your identity to be public for this peer review?** For information about this choice, including consent withdrawal, please see our Privacy Policy .

Reviewer #2: **Yes: ** Dr. Aruna Kumar Barick

Reviewer #3: No

---

## [Editor Report · Acceptance letter]

PONE-D-24-53055R1

PLOS ONE

Dear Dr. Cao,

I'm pleased to inform you that your manuscript has been deemed suitable for publication in PLOS ONE. Congratulations! Your manuscript is now being handed over to our production team.

Kind regards,

on behalf of

Dr. Anil Kumar Reddy Police

Academic Editor

PLOS ONE